# Bio-Functionalized Chitosan for Bone Tissue Engineering

**DOI:** 10.3390/ijms22115916

**Published:** 2021-05-31

**Authors:** Paola Brun, Annj Zamuner, Chiara Battocchio, Leonardo Cassari, Martina Todesco, Valerio Graziani, Giovanna Iucci, Martina Marsotto, Luca Tortora, Valeria Secchi, Monica Dettin

**Affiliations:** 1Department of Molecular Science, University of Padua, Via A. Gabelli 63, 35121 Padua, Italy; 2Department of Industrial Engineering, University of Padua, Via F. Marzolo 9, 35131 Padua, Italy; annj.zamuner@unipd.it (A.Z.); leonardo.cassari@phd.unipd.it (L.C.); martina.todesco@unipd.it (M.T.); monica.dettin@unipd.it (M.D.); 3Department of Science, Roma Tre University of Rome, Via della Vasca Navale 79, 00146 Rome, Italy; chiara.battocchio@uniroma3.it (C.B.); valerio.graziani@roma3.infn.it (V.G.); giovanna.iucci@uniroma3.it (G.I.); martina.marsotto@uniroma3.it (M.M.); luca.tortora@uniroma3.it (L.T.); valeria.secchi@uniroma3.it (V.S.)

**Keywords:** chitosan, functionalization, Chit-HVP, Chit-RGD, XPS, NEXAFS, h-osteoblasts

## Abstract

Hybrid biomaterials allow for the improvement of the biological properties of materials and have been successfully used for implantology in medical applications. The covalent and selective functionalization of materials with bioactive peptides provides favorable results in tissue engineering by supporting cell attachment to the biomaterial through biochemical cues and interaction with membrane receptors. Since the functionalization with bioactive peptides may alter the chemical and physical properties of the biomaterials, in this study we characterized the biological responses of differently functionalized chitosan analogs. Chitosan analogs were produced through the reaction of GRGDSPK (RGD) or FRHRNRKGY (HVP) sequences, both carrying an aldehyde-terminal group, to chitosan. The bio-functionalized polysaccharides, pure or “diluted” with chitosan, were chemically characterized in depth and evaluated for their antimicrobial activities and biocompatibility toward human primary osteoblast cells. The results obtained indicate that the bio-functionalization of chitosan increases human-osteoblast adhesion (*p* < 0.005) and proliferation (*p* < 0.005) as compared with chitosan. Overall, the 1:1 mixture of HVP functionalized-chitosan:chitosan is the best compromise between preserving the antibacterial properties of the material and supporting osteoblast differentiation and calcium deposition (*p* < 0.005 vs. RGD). In conclusion, our results reported that a selected concentration of HVP supported the biomimetic potential of functionalized chitosan better than RGD and preserved the antibacterial properties of chitosan.

## 1. Introduction

Natural biopolymers have attracted more and more attention as tissue engineering scaffolds because of their inherent biophysical and biochemical properties, such as renewability, biocompatibility, biodegradability, and targeting ability. Biopolymers present many advantages over synthetic material-based scaffolds in terms of half-life, stability, safety, and ease of manufacture. Among them, polysaccharide-based scaffolds are non-toxic, non-immunogenic, and display good targeting ability and biocompatibility. They can undergo in vivo enzymatic digestion generating metabolites characterized by low toxicity, too [1]. Moreover, functional groups of the polysaccharide backbone serve as anchoring sites for chemical modifications, generating versatile scaffolds of great significance in the biomedical field [2].

Chitosan is a natural polysaccharide composed of glucosamine and *N*-acetyl-glucosamine, produced by deacetylation of chitin. The amine groups in the glucosidic residue are positively charged at physiological pH. The presence of amino groups assures reactive sites for conjugation. Chitosan is insoluble in water at neutral pH and soluble in acidic conditions when the amino groups are protonated. As it is non-toxic, biodegradable, and biocompatible, the Food and Drug Administration has already approved chitosan for use in wound dressings. Several papers reported on chitosan and its potential application in biomedical fields, including wound healing, tissue regeneration, and drug delivery [3,4]. Indeed, because of its hydrophilic surface, biocompatibility, and biodegradability, chitosan has found significant application in promoting cell adhesion, proliferation, and differentiation [5]. Since chitosan has poor mechanical properties, hybrid materials and copolymers have been evaluated [5]. Sequences of short amino acids of the extracellular matrix (ECM) proteins have been used to guide interactions with specific cell receptors and improve the biological targets of biopolymers by inducing selected cellular responses. Indeed, developing hybrid biomaterials by including peptide sequences into biopolymers is an attractive alternative to confer functional cell-biomaterial interaction. The amino acid sequence RGD (Arginine-Glycine-Aspartic Acid) is a well-known tripeptide that induces cell adhesion and migration by interacting with integrin family receptors. The conjugation of GRGDS to carboxymethyl-trimethyl chitosan, proposed by Hansson A et al. [5], produced a scaffold for dermal healing that increases the initial adhesion 3–5 times and the cell spreading 12-fold compared with unfunctionalized scaffolds. In the article of Wang C et al. [6], a cyclic RGD peptide was conjugated to chitosan by a thiolation reaction and a cross-linking agent and used in addition to graphene oxide in drug delivery applications. Because of its osteoconductive properties and structural homology to proteoglycans of extracellular matrices, chitosan has been proposed as a coating in bone implants or a component in composite materials [7]. So far, experimental studies have reported that alloy surfaces, layer assembly, and the degree of deacetylation n chitosan affect the promotion of osseointegration, bone remodeling, and delivery of growth factors at the implant-tissue interface [8].

In addition to its advantages in tissue regeneration and biomedical applications, chitosan is endowed with intrinsic antimicrobial properties with no toxic effects on host cells [9]. Despite considerable progress, orthopedic device-related infections are clinically relevant problems whose incidence is estimated to be in the range of 0.5–5% for total joint replacements [10]. Infections of endosseous implants are related to pathogenic bacteria usually localized in the skin, such as *Staphylococcus aureus*, that contaminate the wound during a surgical procedure and are attracted by nutrients concentrated in the foreign biomaterial surface of the implant [11]. Adhesion to surface and proliferation induce the formation of bacterial aggregates, namely biofilms, that are highly resistant to antibiotics and human immune responses [12]. The antimicrobial activity of chitosan relies on positive charges of its protonated amino groups, which interact with molecules at the surfaces of bacteria and fungi, resulting in the leakage of intracellular content and cell death [13]. Therefore, the degree of acetylation, molecular weight, and conjugation with functional groups is critical in maintaining chitosan’s antimicrobial properties. Moreover, it has been reported that solid chitosan (i.e., nanoparticles or hydrogels) has a negligible antibacterial effect as compared with soluble chitosan, since nanoparticles expose a reduced number of positive charge available for the binding with the bacterial surface [14].

In this work, the effort to confer new and effective bioactive properties to chitosan through the covalent conjugation with cell adhesive peptides was carried out. The chemistry here proposed for selective covalent conjugation consists of Schiff bases production, successively reduced between chitosan amino groups and aldehyde groups appositely inserted in the peptide backbone during solid phase peptide synthesis. In addition to the RGD peptide, the conjugation with an adhesive peptide named HVP, mapped on the human vitronectin and able to interact with osteoblasts through a proteoglycan-mediated mechanism, was investigated [15,16]. Indeed, previous studies demonstrated that the HVP peptide selects the attachment and supports the differentiation of osteoblasts while promoting cellular migration [17]. The functionalized chitosan matrices (named Chit-RGD and Chit-HVP) have been assessed as scaffolds for osteoblast adhesion, growth, and differentiation. The antibacterial properties have been tested, taking into account the substitutions induced by derivatization. The chemical characterization of Chit-RGD and Chit-HVP was performed through FT-IR, XPS, and NEXAFS analyses.

## 2. Results and Discussion

### 2.1. Chemical Composition and Biomolecule Stability: X-Ray Photoelectron Spectroscopy Investigation

For all samples, SR-induced XPS measurements were performed at the C1s, N1s, and O1s core levels. C1s, O1s, and N1s spectra measured for RGD and Chit-RGD are shown in Figure 1. The RGD C1s core level spectrum can be resolved by the curve-fitting analysis into four components, associated with chemically inequivalent carbon and labeled 1−4 in Figure 1. Peak 1 (BE = 285.0 eV) is attributed to aliphatic (C−C) carbons of the side chains and partially to carbon contamination; peak 2 (286.5 eV) is related to O=C−C−N carbons of the peptide backbone, with a contribution from C−N carbons of the lysine pending groups; peak 3 (288.0 eV) is due to O=C−N peptide carbons and O=C−O^−^ carbons of the aspartate pending group; peak 4 (289.3 eV) is due to O=C−OH carbon sin protonated aspartate and HN=C-NH_2_ carbons in arginine.

The O1s spectra show four-component peaks, labeled 1−4 in Figure 1. Peak 1 (530.0 eV) is assigned to the oxygens of TiO_2_ in the titania substrate; peak 2 (532.1 eV) to O=C oxygens of the peptide backbone and of the carboxylate group of aspartates; peak 3 (533.5eV) to C-O oxygens; peak 4 (535.0 eV) is related to physisorbed water and possibly to C−O−H groups in protonated aspartate. The N1s spectrum results from peak 1 (399.17 eV) due to C=N nitrogens of the arginine pending group, a main peak at 400.4 eV related to peptide nitrogens and peak 3 (402.8 eV), due to protonated nitrogens of the lysine and arginine pending groups [17,18].

The Chit-RGD C1s, O1s, and N1s spectra present a similar shape and the same curve-fitting components of the RGD spectra but different peak relative intensities. For example, the intensity of peak 2 in the C1s spectrum of Chit-RGD shows an evident increase in comparison to peak 3 and a slight shift to higher BE (286.7 eV). Both effects are related to the contributions from C−O carbons of chitosan located at 286.6 eV [19].

Some changes are also present in the O1s spectrum. Peak 1, related to the titania substrate, decreases because of the presence of the chitosan film; peak 3 increases in intensity with respect to peak 2 due to contributions from O−C oxygens of chitosan [20].

Similar results were obtained by the analysis and comparisons of the C1s, O1s, and N1s spectra of HVP and Chit-HVP in Figure 2.

Appendix A lists the Binding Energy (BE), Full Width Half Maxima (FWHM), experimental atomic ratios, and assignments determined from the XPS spectra analysis of all four samples.

The SR-induced XPS data analysis confirms the successful anchoring of both peptides to titania surfaces, as well as their chemical stability. What is more, the substantial modifications arising in the C1s and O1s spectra of the peptides functionalized with chitosan are indicative of a successful peptide-chitosan conjugation, as expected from the synthetic path described in Section 3.2.2.

### 2.2. Molecular Structure and Organization: Near Edge X-ray Absorption Fine Structure Studies

First, we will analyze NEXAFS C and N K-edge spectra collected at the magic incidence angle (54.7° of incidence of the X-ray radiation on the sample surface, ensuring that no dichroic contributions arise from the sample’s spatial orientation) on RGD, the complex Chit-RGD, HVP, and the complex Chit-HVP.

The carbon K-edge spectra of RGD and Chit-RGD are reported in Figure 3A. Those of HVP and Chit-HVP are shown in Figure 3B. The feature at about 288.7 eV is associated with a C 1s→π* transition of a C=O molecular orbital, the right shoulder at about 289.0 eV to a C 1s→π* transition of a C=N molecular orbital in the arginine pending group, and the left shoulder at about 288.0 eV to the histidine pending group just in the case of HVP and Chit-HVP [21]. The feature at 292 eV can be assigned to 1s→σ* transitions by the C−N and C-O molecular groups present in the amino acid pending groups and in chitosan in the case of Chit-RGD and Chit-HVP, so that this feature appears more intense in the Chit-RGD and Chit-HVP spectra. Additional features at 294 and 303 eV can be associated with 1s→σ* transitions by C−C and C=O molecular groups, respectively [22].

As for N K-edge spectra (reported in Figure 4A,B), all samples show a sharp peak at 402 eV assigned to N 1s→π* transitions of the peptide bonds, one peak at 399.5eV due to N 1s→π* transitions of C=N arginine [23] and two broadbands at 406 and 413 eV to N 1s→σ*N−H and N 1s→σ*N−C resonances, respectively [22]. The spectra of HVP and Chit-HVP also show a peak at 400.4 eV related to C=N histidine [24].

To evaluate if a molecular preferential orientation occurs, we performed NEXAFS measurements at both C K-edge and N K-edge by varying the photon beam incidence angle on the sample surface from grazing (20°) to normal (90°). To highlight the most indicative dichroic effects, in Figure 5, angular dependent N K-edge spectra collected on RGD (a), HVP (b), Chit-RGD (c), and Chit-HVP (d) anchored onto titania surfaces are reported. As can be seen, HVP and Chit-HVP show a strong dichroic effect in the π* region; the ratio between the peak intensities at normal (90°) and grazing (20°) incidence, determined for the selected resonance by peak fitting of the experimental data, can be inserted in the equation reported by Stöhr [25] for the 3-fold or higher symmetry substrates, assuming the polarization factor *p* = 0.95, allowing to calculate the average tilt angle between the π* vector orbital for the C=O bond and the normal to the surface. For HVP (c) and Chit-HVP (d), the tilt angle of the peptide chains is about 65.5° and 67.5°, respectively. Interestingly, no dichroic effects or minimal effects are observed for RGD and Chit-RGD.

### 2.3. FTIR Spectroscopy

FTIR spectra of pristine peptides HVP and RGD are shown in Figure 6. In the spectra of the samples, at high wavenumbers, the N-H stretching vibrations produce the broad peak at about 3300 cm^−1^, while the two sharp peaks localized at 2930 and 2850 cm^−1^ can be ascribed to C-H stretching vibrations. At lower wavenumbers, the spectra evidence peaks related to the peptide bond, namely the C=O stretching vibration, or amide (I), located at 1660–1630 cm^−1^ and the N-H bending band, or amide (II), located at 1550–1530 cm^−1^ for both peptides. The position of amide (I) can be used to determine the peptide’s secondary structure; a wavenumber of about 1660 cm^−1^ can be ascribed to an alpha-helix type conformation of the peptide backbone for most proteins and peptides, according to the literature [26]. The position of amide (II) is also in accordance with this evidence [27]. However, contributions to the IR spectra could also be due to the amino acids pending groups; the peptide sequence is GRGDSPK for RGD and FRHRNRKGY for HVP. In particular, the HVP peptide contains three arginine residues; however, the main contribution to the IR spectrum is the C=N stretching band of the guanidine moiety, located at about 1640 cm^−1^, and therefore superimposed with the main amide (I) band [28]. The infrared spectra of the Chit-HVP and Chit-RGD assemblies, shown at the bottom of Figure 6, result from contributions of chitosan and of the immobilized peptides. In the chitosan spectrum [29,30,31], the most intense peaks are the broadband located at 3550 cm^−1^ and the intense peak located at 1080 cm^−1^, attributed to O-H and C-O stretching vibrations, respectively. These peaks are evident in the Chit-HVP and Chit-RGD spectra of Figure 6. Contributions due to O-H bending vibrations around 1400 cm^−1^ are also evident in both spectra. For sample Chit-HVP, the position of the amide (I) and amide (II) bands reproduces the same features in the spectrum of the pristine HVP peptide. It is worth noticing that, since chitosan is obtained by the partial hydrolysis of chitin, weak residual amide (I) and amide (II) bands can be found in the spectrum of chitosan due to the incomplete hydrolysis of the amide functions. However, the position and intensity of the peaks in the Chit-HVP spectrum are a clear indication of successful peptide grafting to chitosan and of the retention of the peptide’s secondary structure after immobilization.

For the Chit-RGD assembly, the situation appears more complex. The two bands are less intense and appear shifted with respect to the pristine RGD spectrum. The amide (I) band appears as a shoulder at about 1640 cm^−1^, on the high wavenumber side of the amide (I) band, which seems to be shifted to about 1580 cm^−1^. These data could indicate that a lower peptide amount has been grafted to chitosan, in agreement with the XPS results.

### 2.4. Functionalization of Chitosan with Biomimetic Peptides Affects Human Osteoblast Survival and Adhesion

The biomimetic properties of chitosan matrices functionalized with HVP and RGD peptides were assessed in primary human osteoblast cells by the MTT test. As reported in Figure 7A, following 2 h in culture, chitosan matrices functionalized with only HVP (Chit-HVP 100% in concentration) reported an increased number of attached osteoblast cells compared with non-functionalized chitosan (Chit). Chitosan functionalized with RGD 100% (Chit-RGD 100%) did not report a significant increase in cell adhesion compared with Chit-HVP 100% or Chit.

Different combinations of HVP and RGD with Chit or the two peptide concentrations were also tested to identify the best-performing blend. After 2 h in culture, we observed that the number of viable cells was reduced with the decreasing concentrations of the biomimetic peptides used for functionalized chitosan (Figure 7B). Human osteoblast cells preferentially attach to, and survive in, Chit-HVP 100%. Considering that HVP and RGD interact with the same integrin receptor family on cells [5,15,16], we combined HVP and RGD at different concentrations to functionalized chitosan matrices. As reported in Figure 7C, the addition of 25% HVP improved the biomimetic properties of Chit-RGD as reported by the increased number of viable cells in Chit-HVP25%RGD75% (*p <* 0.05 vs. Chit-RGD 100%).

On the other hand, Chit-HVP 100% is the best performing functionalization, since the replacement of 25% of HVP with RGD significantly worsens the ability to attract osteoblast cells (Figure 7C). Overall, by testing different combinations of peptide concentrations, we did not observe an increase in the number of viable cells compared with Chit-HVP 100%, ruling out a possible synergistic effect of the two adhesive sequences. Since it has been reported that a positive surface charge, the pH, and the molecular weight of chitosan impact its stability and biological activity [5], our data reveal that the molecular structure of Chit-HVP (Figure 4 and Figure 5) supports the biomimetic potential of chitosan better than RGD functionalization.

The adhesion of osteoblast cells on differentially functionalized chitosan matrices was evaluated by immunocytochemistry, analyzing the expression and distribution of the smooth muscle actin (α-SMA), a marker of cytoplasmic fibers involved in stress tensor and mechanotransduction during cell adhesion [32,33]. As reported in Figure 8, after 24 h in culture, osteoblast cells adhered to the chitosan matrix and reported a blunt expression of α-SMA compared with cells cultured on unfunctionalized glass coverslips. Osteoblast cells cultured on Chit-HVP 100% or Chit-HVP 50% are elongated and pointed on both ends, exhibiting a spindle shape. Moreover, in cells cultured on Chit-HVP 100%, actin fibers are structurally organized in a filament network, and the expression of α-SMA increased. Cells cultured on Chit- RGD 100% or Chit-RGD 50% reported an elongated shape, but actin fibers accumulate in the cytoplasm, resulting in a less organized distribution of α-SMA.

### 2.5. HVP Functionalization of Chitosan Increases Proliferation of Human Osteoblast Cells

Cell proliferation was evaluated using the intracellular dye CFSE and a cytofluorimetric analysis on human osteoblast cells cultured 72 h on different functionalized chitosan matrices. As reported in Figure 9, in cells cultured on a plastic culture surface or non-functionalized chitosan matrices, the fluorescent intensity relative to cell proliferation was recorded in 3.17 ± 1% and 4.3 ± 1.1% of cells, respectively. Cells cultured on Chit-HVP 100% reported a significant increase in cell proliferation (*p* < 0.05) compared with Chit. However, cell proliferation decreased in Chit-HVP 50% to levels comparable to those of chitosan matrices functionalized with RGD.

Several studies reported chitosan ability in sustaining osteoblast proliferation when chitosan is used as a coating in titanium implants or alone in the form of devices for bone regeneration. Indeed, chitosan nanofiber scaffolds increased DNA replication and cell proliferation in a time-dependent manner, up to 5 days [34]. In our study, cell proliferation was evaluated following 72 h of culture, the time at which we observed the maximal proliferative effect in a preliminary set of experiments. Even if our study did not consider chitosan organized in nanofibers, unlike chitosan or RGD, HVP functionalization was the only one able to augment osteoblast proliferation, a critical event in bone tissue regeneration.

### 2.6. Chitosan Functionalization with HVP Supports Human Osteoblast Differentiation

To further evaluate the role of functionalization of chitosan matrices in sustaining the differentiation of primary human osteoblast cells, the expression of genes involved in bone formation was assessed by a quantitative RT-PCR. As reported in Figure 10, cells cultured on Chit-HVP 100% and Chit-HVP 50% increased the mRNA specific transcript levels of *Spp1* gene-coding osteopontin involved in bone homeostasis [35]. At the same time, Chit-HVP increased the expression of *Runx2* mRNA (Figure 10), a key transcription factor associated with increased levels of osteocalcin and sialoprotein involved in osteoblast differentiation [35].

To confirm osteoblast differentiation, we cultured cells on functionalized chitosan matrices for seven days. The cells were then stained with Alizarin red. As reported in Figure 11, osteoblast cells cultured for 7 days on Chit-HVP 100% and Chit-HVP 50% reported strong signals relative to Alizarin staining, demonstrating increased calcium deposition. Calcium deposition was evident to a less extent in cells cultured on matrices functionalized with RGD. In a parallel set of experiments, the dye was extracted from cells, and optical density (O.D.) was determined to quantify calcium deposition. As reported in Figure 11B), we observed an increase in O.D. values in cells cultured in Chit-HVP, confirming the increased formation of calcium deposition.

### 2.7. Chitosan Functionalization with Biomimetic Peptides Reduces the Antibacterial Activity

As reported, the antimicrobial activity of chitosan is linked with the conjugation to functional groups and the degree of acetylation [13]. In the context of endosseous implants, devices naturally endowed with antimicrobial effects confer important advantages.

Therefore, we evaluated whether the functionalization of chitosan with HVP or RGD alters the antimicrobial activity of chitosan matrices against the biofilm of *S. aureus*. To this aim, cultures of *S. aureus* were grown for 48 h on different functionalized chitosan matrices and then stained using a LIVE/DEAD BacLight Bacterial Viability Kit. As reported in Figure 12A, the antibacterial effect of chitosan was confirmed against *S. aureus*, as very few bacterial cells were retrieved compared to the glass surface. Functionalization with HVP or RGD reduced chitosan’s antibacterial effect, as more bacterial cells were visualized in Chit-RGD 100 and Chit-RGD 50 or Chit-HVP 100 and Chit-HVP 50 as compared to the unfunctionalized chitosan. Live and dead bacteria were enumerated and, considering the different number of bacterial cells in functionalized surfaces, viable *S. aureus* increased as the concentrations of the peptides increased (Figure 12B), suggesting that structural characteristics of functionalized chitosan guide the antibacterial effect. Generally, chitosan reported a stronger bactericidal effect against Gram-positive bacteria than against Gram-negative bacteria [36]. In chitosan functionalized with HVP, we should expect a similar trend. However, by tuning the concentration of HVP, and thus changing the chemical charges at the surface of Chit-HVP, we envisage the possibility to improve the antimicrobial effects of chitosan against Gram-negative bacteria.

Overall, our data indicate that chitosan bio-functionalization with HVP confers beneficial features on endosseous devices but also lead to the loss of the natural antibacterial effect of chitosan. The reported results showed the different biomimetic properties of the two adhesive sequences resulting in different yields of functionalization, as demonstrated by XPS and FT-IR data, and correlated to different bioactivity of synthetic peptides. On the other hand, it is well known that the cellular response depends on a peculiar pattern and concentration of peptides that need to be optimized. Actually, several circulating proteases are responsible for biomimetic peptides’ degradation in vivo, thus complicating the performance of functionalized biomaterials. Therefore, as in vitro experiments do not take into account the in vivo variables, more tunable systems should be set to calibrate the concentration of biomimetic peptides on hybrid biomaterials. Indeed, given the results reported in this study, the best performing matrix for bone tissue application results in Chit-HVP 100%. However, the analysis of cell and bacterial adhesion reveals that Chit-HVP 50% should preserve both the antibacterial feature of the biomaterial and the beneficial cellular features.

## 3. Materials and Methods

### 3.1. Materials

Acetic acid (AcOH), NaOH, Methanol (MeOH), trifluoroacetic acid (TFA), Triethoxysilane (TES), *N*,*N*-dimethylformamide (DMF), and Pyridoxal 5-phosphate (PLP) were purchased from Sigma-Aldrich. 2-(1H-benzotriazol-1-il)-1,1,3,3-tetramethyluronium exafluorophosphate (HBTU), 1-hydroxybenzotrazole (HOBt), H-Phe Nova Syn Tg resin, and all Fmoc-protected amino acids were purchased from Merck (Darmstadt, Germany). *N*-methyl-2-pyrrolidone (NMP), dichloromethane (DCM), *N*,*N*-Diisopropylethylamine (DIEA), and piperidine were from Biosolve (Leenderweg, Valkenswaard, The Netherlands). Ethanol (EtOH) was provided by VWR Chemicals (Milan, Italy). Chitosan 70/1000 was purchased from Heppe Medical Chitosan GmbH (Halle, Germany).

### 3.2. Methods

#### 3.2.1. Peptide Synthesis

(1)GRGDSPK-Aldehyde

The GRGDSPK peptide, named RGD for simplicity (sequence H-Gly-Arg-Gly-Asp-Ser-Pro-Lys-NH_2_) was synthesized with Fmoc Chemistry and a solid phase strategy using a Syro I synthetizer (MultiSynTech, Witten, Germany). Double coupling was carried out for loading. The peptide was cleaved from the resin with TFA:H_2_O:TES (95:2.5:2.5) for 1.5 h under stirring. The peptide was purified by RP-HPLC. The conversion of the N terminal group NH_2_-CH_2_- into an aldehyde group was achieved by treating the peptide with 20 molar equivalents of PLP in a 25 mM phosphate buffer at pH = 6.5 for 18h at 37 °C. Finally, the GRGDSPK-aldehyde was purified by RP-HPLC up to 98% homogeneity (Appendix A). The identity of the target product was determined through mass spectrometry (Appendix A).

(2)HVP-Aldehyde

The HVP-aldehyde peptide (sequence H-Phe-Arg-His-Arg-Asn-Arg-Lys-Gly-Tyr-7-aminoheptanoic acid-Phe-H) was synthesized with Fmoc Chemistry and a solid phase strategy using a Syro I synthesizer (MultiSynTech, Witten, Germany). H-Phe Nova Syn Tg resin (0.2 mmoles/g) was used as solid support. The 7-aminoheptanoic acid, introduced as a spacer, was conjugated using a single coupling, whereas the following residues were introduced with double couplings. The side-chain deprotection was carried out by treating the peptide-resin with 4 mL TFA for 30 min. The resin was washed with DCM and dried under vacuum for 30 min, and then the cleavage of the peptide from the solid support was obtained with 9.9 mL of an AcOH:H_2_O:DCM:MeOH (10:5:63:21) solution treatment for 1h under magnetic stirring. The crude peptide was purified by RP-HPLC and characterized by analytical RP-HPLC and mass spectrometry as reported in the Appendix A.

#### 3.2.2. Functionalization of Chitosan

Chitosan (100 mg) dissolved in 5.5 mL AcOH 0.2M. EtOH (3.75 mL) was added and the pH was adjusted to 5.1 with NaOH 1N. GRGDSPK-aldehyde (11.54 mg) was summed. Finally, 100 mg of NaCNBH_3_ were added. The solution was stirred for 24 h at room temperature. After adjusting the pH to 7 with NaOH 1N, the addition of EtOH caused the product precipitation. Chit-RGD was filtered with gooch G3 and dried under vacuum for 1 h. The same protocol was used for the preparation of Chit-HVP, starting from 14 mg of chitosan and treating with 12.87 mg of HVP-aldehyde.

#### 3.2.3. Chemical Characterization

Samples for chemical analysis were prepared by incubation of TiO_2_/Si(111) surfaces for about 16 h with a mother solution of the selected biomolecules (0.4 mg of Chit-RGD or, respectively, Chit-HVP dissolved in 120 µL of an AcOH 0.2M water solution).

(1)XPS

XPS experiments have been performed at MSB, Materials Science Beamline, at ELETTRA-Sincrotrone Trieste, using an ultra-high vacuum experimental system. MSB is placed at a bending magnet exit and, using a plane grating monochromator, can provide photon energy ranging from ultraviolet to soft X-rays (20 to 1000 eV). The X-ray spectrometer operates at the background pressure of low 10^−10^ mbar, and the emitted photoelectrons are analyzed by a SPECS Phoibos 150 electron energy analyzer.

For SR XPS measurements, photoelectrons have been collected normally to the sample surface, while the radiation impinges with 60° with respect to the sample surface.

All spectra were energy-referenced to the C1s signal of aliphatic carbon located at 285.0 eV. A curve-fitting analysis of the experimental spectra was performed using Gaussian curves as fitting functions. Atomic ratios were calculated from peak intensities using Scofield’s cross-section values.

(2)Near-Edge X-ray Absorption Fine Structure (NEXAFS)

NEXAFS spectra were acquired at the BEAR beamline (bending magnet for emission absorption and reflectivity), at the ELETTRA storage ring. BEAR is installed at the left exit of the 8.1 bending magnet exit. The apparatus is based on a bending magnet as a source and beamline optics delivering photons from 5 eV up to about 1600 eV with a selectable degree of ellipticity. The UHV end station is equipped with a movable hemispherical electron analyzer and a set of photodiodes to collect angle-resolved photoemission spectra, optical reflectivity, and fluorescence yield. In these experiments, we used ammeters to measure the drain current from the sample. C and N K-edge spectra were collected at normal (90°), grazing (20°), and magic (54.7°) incidence angles of the linearly polarized photon beam with respect to the sample surface. In addition, our carbon and nitrogen K-edge spectra have been further calibrated using the resonance at 288.70 eV, assigned to the C=O 1s→π* transition, and the resonance at 402.00 eV, assigned to the 1s→π* transition of the peptide bonds, respectively. The raw C and N K-edge NEXAFS spectra were normalized to the incident photon flux by dividing the sample spectrum by the spectrum collected on a freshly sputtered gold surface. The spectra were then normalized by subtracting a straight line that fits the part of the spectrum below the edge and setting to 1 the value at 330.00 and 430.00 eV for carbon and nitrogen, respectively.

(3)FT-IR

A Continuum FT-IR microscope mounted on a Nicolet iS50 FT–IR spectrometer (ThermoFisher^®^, Waltham, MA, USA) working with a liquid-nitrogen-cooled MCT (mercury-cadmium telluride) detector was employed for acquiring FT-IR spectra of the investigated samples. The spectra were collected in μ–ATR mode employing a Ge tip, in the region of 400–4000 cm^−1^, with a resolution 8 cm^−1^ and 256 scans per spectrum. The optical bench apertures were set to 100 and the gain to 8.

#### 3.2.4. In Vitro ASSAYS

(1)Human Osteoblast Cells’ Isolation and Culture

Human (h) osteoblast cells were obtained from explants of cortical mandible bone collected during a surgical procedure. After collection, the bone fragments were cultured at 37 °C in 5% CO_2_ and 95% humidity in a complete medium (DMEM supplemented with 10% *v*/*v* heat-inactivated fetal bovine serum, and penicillin-streptomycin solution, all provided by ThermoFischer) and incubated until the cells migrated from the bone fragments. At the cell confluence, the bone fragments were removed and the cells were detached using trypsin EDTA (Gibco) and cultured in a complete medium supplemented with 50 mg/mL ascorbic acid, 10 nM dexamethasone, and 10 mM β-glycerophosphate (all purchased from Merck). The osteoblast phenotype was confirmed by the von Kossa staining. The study was approved by the Ethical Committee of the University Hospital of Padova. The patients were informed of the study’s aims and protocol and gave their written informed consent.

(2)Cell Adhesion

Osteoblast cells were seeded and cultured for 2 h on different functionalized chitosan matrices. Cell viability was assessed using the MTT (3-(4,5-dimethylthia-zole-2-yl)-2,5-diphenyl tetrazoliumbromide; Merck) assay. Briefly, at the end of incubation, the cells were rinsed three times with PBS to remove non-adherent cells and then incubated with MTT (5 mg/mL in 100 μL final volume) at 37 °C for 4 h. The reaction was stopped by adding 0.01 N HCl in 10% *v*/*v* SDS. The cells were quantified by setting a standard curve for each experiment, obtained by seeding a known number of h-osteoblast cells. The absorbance of cell lysates was recorded at 620 nm.

(3)Immunofluorescence

Human osteoblast cells were cultured for 24 h on chitosan matrices layered on glass coverslips. The cells were fixed in 4% *w*/*v* paraformaldehyde (PFA) for 10 min and then washed three times (5 min each) in PBS. The cells were permeabilized with 0.1% Triton X-100, and nonspecific binding sites were then blocked by incubation with 2% bovine serum albumin (BSA) in PBS for 30 min. The cells were incubated with an anti-αSmooth Muscle Actin (α-SMA) antibody conjugated with tetramethylrhodamine B isothiocyanate (TRITC; 50 mg/mL, Merck). For nuclear counterstaining, the samples were incubated with TOTO-3 conjugated with Alexa Fluor 647. After extensive washing, the samples were mounted and analyzed using a Leica TCSNT/SP2 confocal microscope. The images were digitally stored using the Leica software.

(4)Proliferation Assay

The cells were incubated at 37 °C for 10 min in a prewarmed PBS containing 0.1% *v*/*v* BSA and a 25 mM carboxyfluorescein diacetate succinimidyl ester (CFSE, Molecular Probe, Invitrogen). Staining was quenched by adding five volumes of ice-cold culture media. The cells were then washed, counted using Trypan blue, seeded at 8 × 10^4^ cells/mL, and incubated in a fresh culture medium at 37 °C for 72 h. Cell proliferation was evaluated by the partitioning of a fluorescent dye between daughter cells using a BD FACS-Calibur flow cytometer.

(5)Quantitative Real-Time Polymerase Chain Reaction

Specific mRNA transcript levels coding human Secreted Phosphoprotein 1 (*Spp1*) and Runt-related transcription factor 2 (*Runx2*) were quantified in osteoblast cells cultured 24 h on differently functionalized chitosan matrices. At the end of incubation, the total RNA was extracted using the EZN.A lysis buffer (Total R.N.A. Kit I (Omega Bio-Tek, Milan, Italy). Contaminating DNA was removed by DNase I treatment (Omega Bio-Tek). cDNA synthesis and subsequent polymerization were performed in a one-step using the iTaq Universal SYBR Green One-Step Kit (Bio-Rad). The reaction mixture contained a 200 nM forward primer, 200 nM reverse primer, iTaq universal SyBR Green reaction mix, and iScript reverse transcriptase, and a 200 ng total RNA Real-time PCR was performed using the ABI PRISM 7700 Sequence Detection System (Applied Biosystems, Milan, Italy). Data were quantified by the ΔΔC_T_ method using hGAPDH as the reference gene. Target and reference genes were amplified with efficiencies near 100%. The oligonucleotides used for PCR are listed in Table 1.

(6)Alizarin Staining

Osteoblast cells cultured on differently functionalized chitosan matrices were washed with PBS and fixed using 10% PFA for 30 min. The cells were then stained with 40 mM Alizarin red (pH 4.2) for 40 min in the dark at room temperature. The cells were washed in distilled water and mounted. The images were obtained using a Leica microscope equipped with a digital camera. In a parallel set of experiments, cells fixed in PFA were incubated at −20 °C for 30 min and then lysed in acetic acid 10% *v*/*v*. The samples were incubated at 85 °C for 10 min and centrifuged, and the supernatants’ pH was neutralized before reading the absorbance of Alizarin red at 405 nm using a microplate reading (Tecan).

(7)Bacterial Cultures and Biofilm Staining

*Staphylococcus aureus* was purchased from CCUG (Culture Collection University of Gothenburg, Gothenburg, Sweden) and maintained in a Trypticase Soy (TS) agar or broth. Bacterial cells were harvested from overnight cultures, and 10^2^ colony forming units (CFU)/mL were cultured on differently functionalized chitosan matrices layered on glass coverslips. The bacteria were cultured in a total volume of 2 mL of TS broth for 48 h at 37 °C without agitation. At the end of incubation, the glasses were washed in 0.85% *w*/*v* NaCl to remove loosely bound bacterial cells. The biofilms were stained with the LIVE/DEAD BacLightTM Bacterial Viability Kit (Thermo Fisher, Milan, Italy) for 15 min at room temperature in the dark. Briefly, the samples were incubated with 5 μM SYTO9 (a green fluorescent nucleic acid dye labeling both live and dead bacteria) and 30 μM propidium iodide, penetrating only bacteria with damaged membranes. Unlabeled dyes were removed by washing, and the samples were visualized using confocal microscopy. Data were analyzed using ImageJ, and the ratio of live/dead bacteria (i.e., bacteria with a high propidium iodide fluorescence and a low SYTO9 green fluorescence) was recorded.

(8)Statistical Analysis

Biological data are reported as mean ± standard error. The statistical analysis was performed using the One-way ANOVA test followed by Bonferroni’s multicomparison test, using Graph-Pad Prism 3.03. *p*-values < 0.05 were considered statistically significant.

## 4. Conclusions

Our results reported that the molecular structure of Chit-HVP supported the biomimetic potential of chitosan better than RGD functionalization. In fact, after 2 h of culture, Chit-HVP resulted in the best performing functionalization in osteoblast cell adhesion, whereas even a 25% replacement of Chit-HVP content with Chit-RGD significantly decreased cell adhesion. Furthermore, at 72 h of incubation, osteoblasts cultured on Chit-HVP reported a significant increase in cell proliferation compared with Chit, and cell proliferation decreased with the decreasing percentage of HVP content. Chit-HVP 100% and Chit-HVP 50% supported the adhesion of osteoblasts, the formation of elongated cell shapes, and increased osteoblast differentiation. Even if chitosan bio-functionalization with HVP conferred beneficial features on endosseous devices, it reduced the antibacterial effect of chitosan against biofilms of *S. aureus*. While Chit-HVP 100% showed the best results in osteoblast adhesion, proliferation, and differentiation, only Chit-HVP 50% preserved the antibacterial effect and supported cell adhesion.

## Figures and Tables

**Figure 1 ijms-22-05916-f001:**
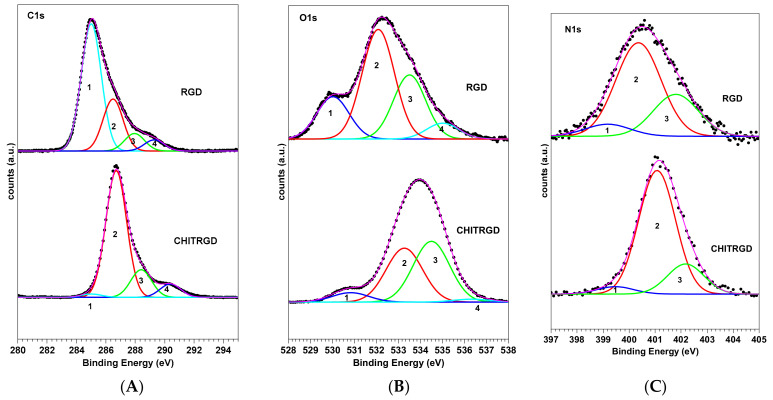
C1s (**A**), O1s (**B**) and N1s (**C**) spectra of RGD peptide and functionalized chitosan matrix Chit-RGD deposited on titania and curve-fitting analysis of the experimental spectra.

**Figure 2 ijms-22-05916-f002:**
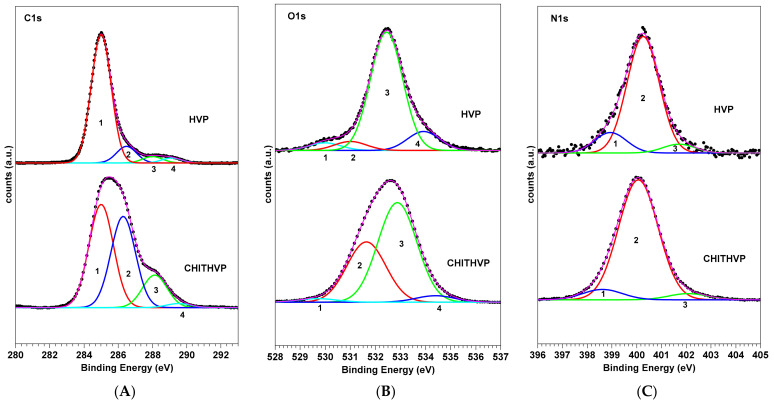
C1s (**A**), O1s (**B**) and N1s (**C**) spectra of HVP peptide and Chit-HVP functionalized chitosan matrix deposited on titania and curve-fitting analysis of the experimental spectra.

**Figure 3 ijms-22-05916-f003:**
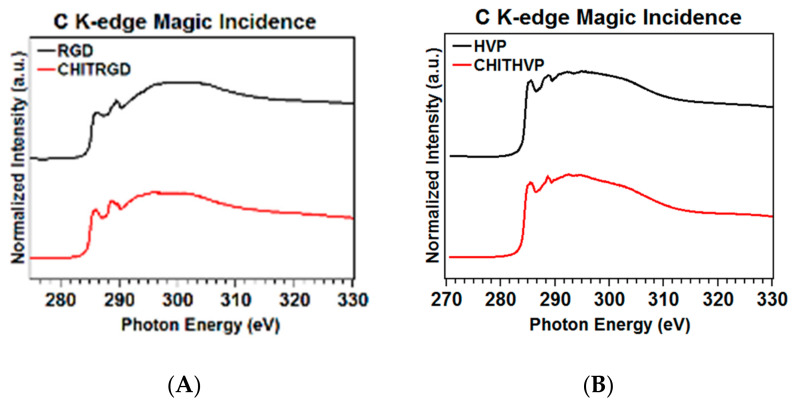
Carbon K-edge spectra of RGD and Chit-RGD (**A**) and of HVP and Chit-HVP (**B**).

**Figure 4 ijms-22-05916-f004:**
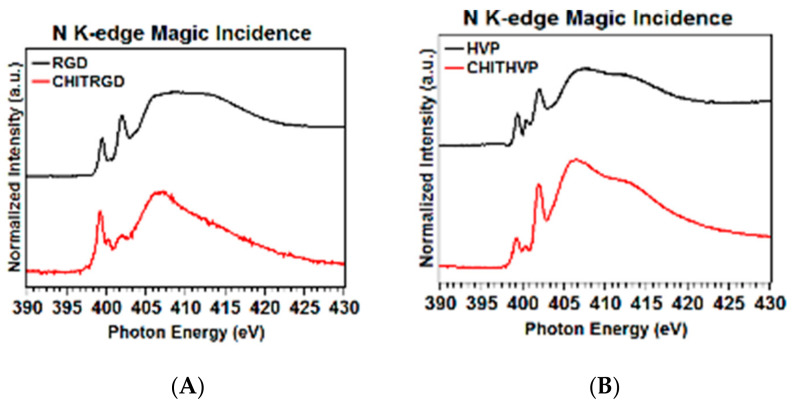
Nitrogen K-edge spectra of RGD and Chit-RGD (**A**) and of HVP and Chit-HVP (**B**).

**Figure 5 ijms-22-05916-f005:**
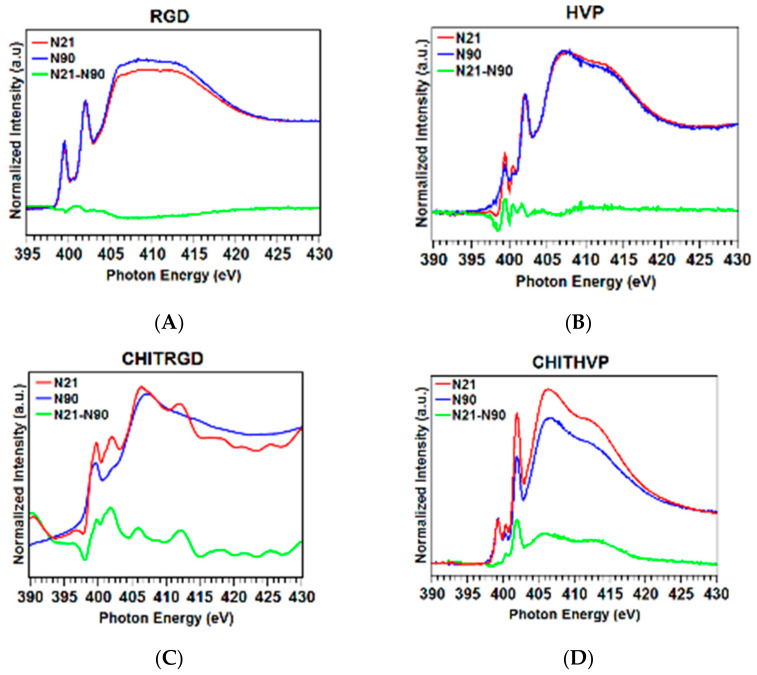
Angular dependent N K-edge NEXAFS spectra collected at normal (90°) and grazing (20°) incidence angles of the impinging X-ray beam on (**A**) RGD; (**B**) HVP; (**C**) Chit-RGD; (**D**) Chit-HVP. The difference spectra (grazing-normal), evidencing dichroic effects, are also shown (green lines).

**Figure 6 ijms-22-05916-f006:**
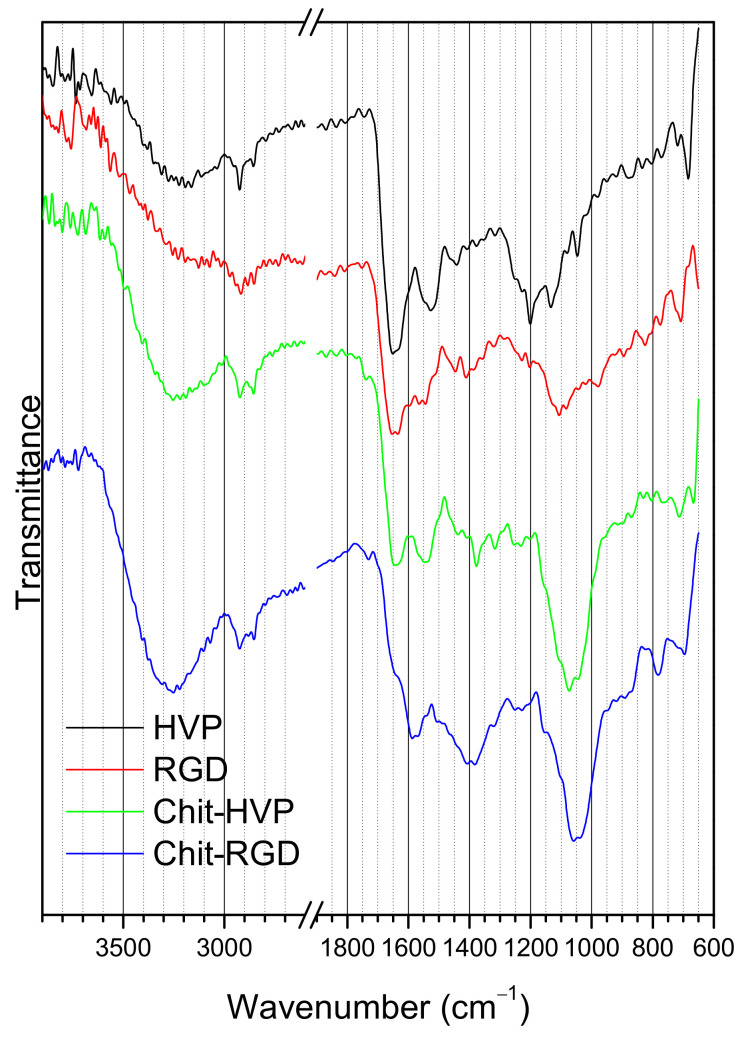
FTIR spectra of the pristine peptides (HVP, RGD) and the corresponding chitosan-peptide assemblies (Chit-HVP, Chit-RGD) in the 3900–2600 and 1900–600 cm^−1^ wavenumber ranges.

**Figure 7 ijms-22-05916-f007:**
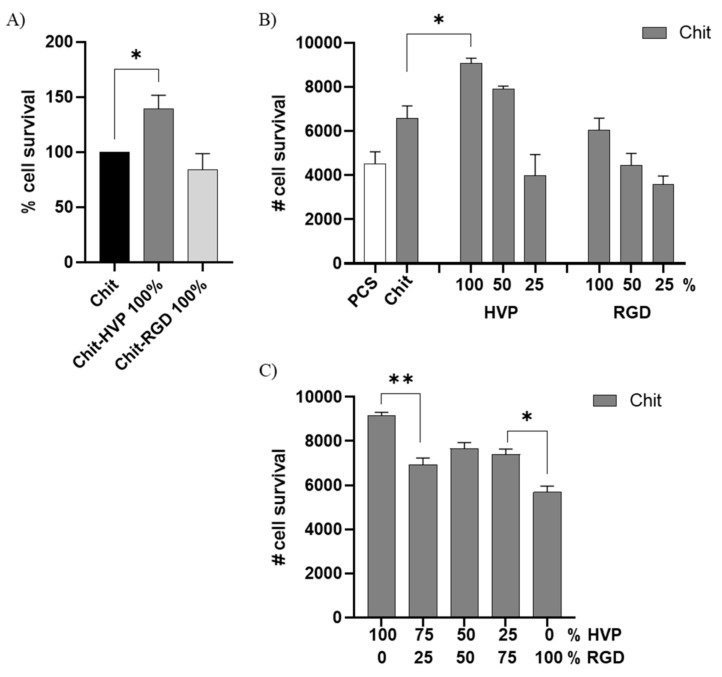
MTT assay performed in primary human osteoblast cells seeded for 2 h on different functionalized chitosan matrices. The cells were quantified by setting a standard curve for each experiment obtained by planting a known number of osteoblast cells. In (**A**), the percentage of cell survival was calculated on the number of cells retrieved in chitosan. (**B**,**C**) number of cells adhering to different functionalized chitosan. Data are reported as mean ± st err of three independent experiments, each performed in duplicate. * indicates *p* < 0.005. ** indicates *p* < 0.001. PCS: plastic culture surface; Chit: chitosan. % indicates the concentration of HVP or RGD.

**Figure 8 ijms-22-05916-f008:**
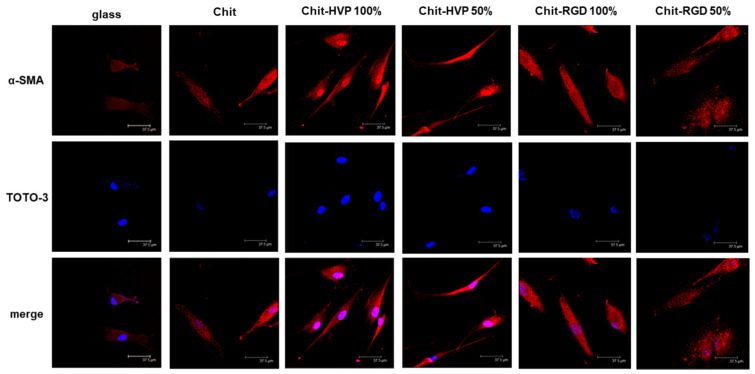
Immunofluorescence on human primary osteoblast cells cultured for 24 h on differently functionalized chitosan matrices. Cells were stained with anti-αSMA antibody (red). Nuclei were stained with TOTO-3 (blue). The images were acquired using a Leica TCSNT/SP2 confocal microscope (Leica Microsystems). Scale bar: 37.5 μm. The images are representative of three independent experiments.

**Figure 9 ijms-22-05916-f009:**
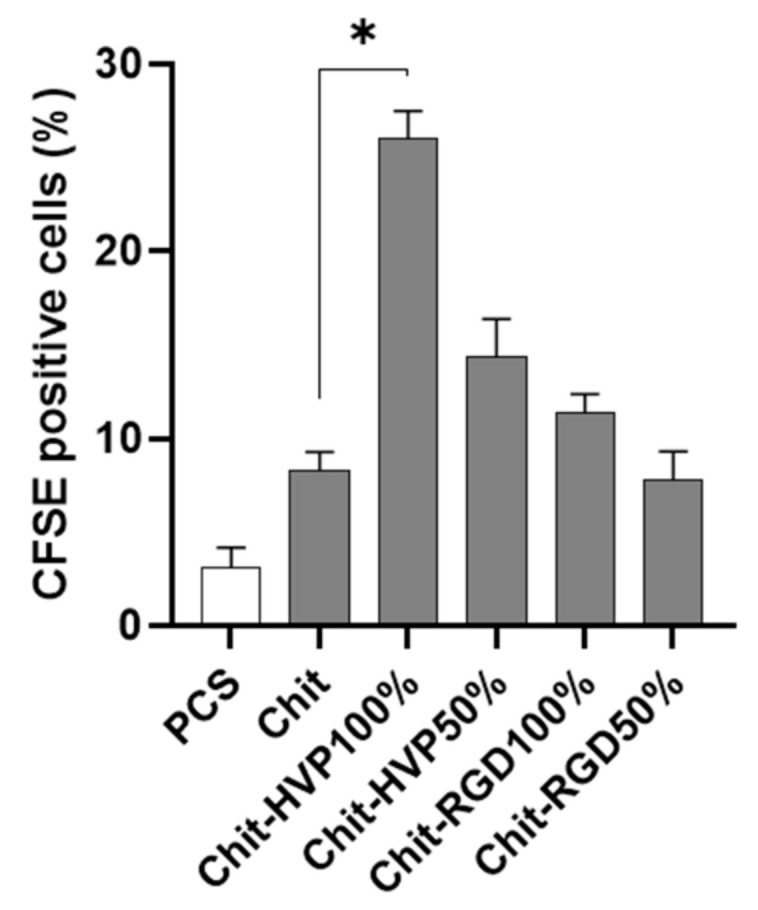
Cell proliferation evaluated in primary human osteoblast cells loaded with CFSE intracellular fluorescent dye. The cells were cultured for 72 h on chitosan matrices and then analyzed by cytofluorimetric analysis. Data are reported as mean ± st err of three independent experiments, each performed in duplicate. * indicates *p* < 0.005. PCS: plastic culture surface; Chit: chitosan. % indicates the concentration of HVP or RGD.

**Figure 10 ijms-22-05916-f010:**
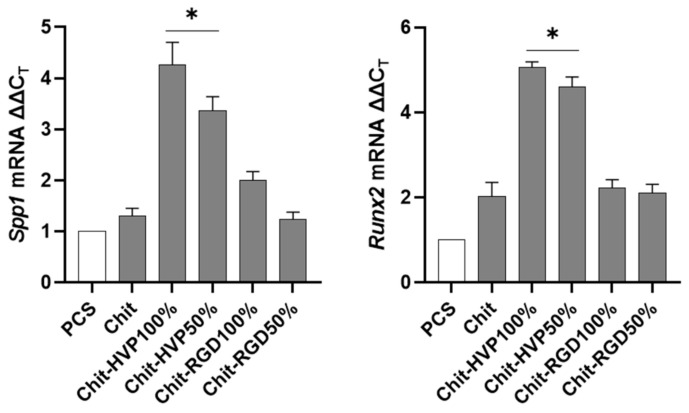
Quantitative RT-PCR performed on human osteoblast cells cultured for 24 h on functionalized chitosan matrices. Data were quantified by the ΔΔC_T_ method using hGAPDH as a reference gene. Data are reported as mean ± st err of three independent experiments, each performed in triplicate. * indicates *p* < 0.005 vs. Chit. PCS: plastic culture surface; Chit: chitosan. % indicates the concentration of HVP or RGD.

**Figure 11 ijms-22-05916-f011:**
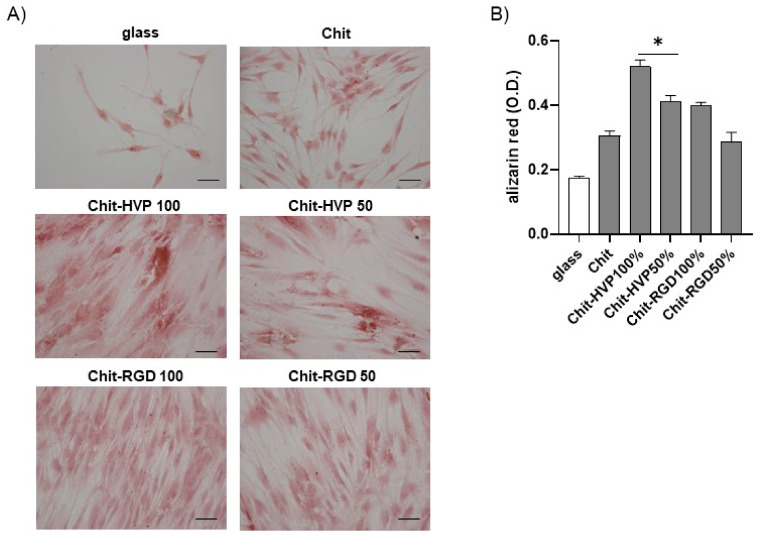
Osteoblast cells were cultured for 7 days on functionalized chitosan matrices and then fixed and stained with Alizarin red to visualize calcium deposition. (**A**) The images were acquired using a Leica microscope equipped with a digital camera. (**B**) The cells were lysed in acetic acid, and the optical density (O.D.) was recorded at 405 nm. Data are reported as the mean ± st err of three independent experiments, each performed in duplicate. * indicates *p* < 0.005 vs. Scale bar: 75 μm. Chit. PCS: plastic culture surface; Chit: chitosan. % indicates the concentration of HVP or RGD.

**Figure 12 ijms-22-05916-f012:**
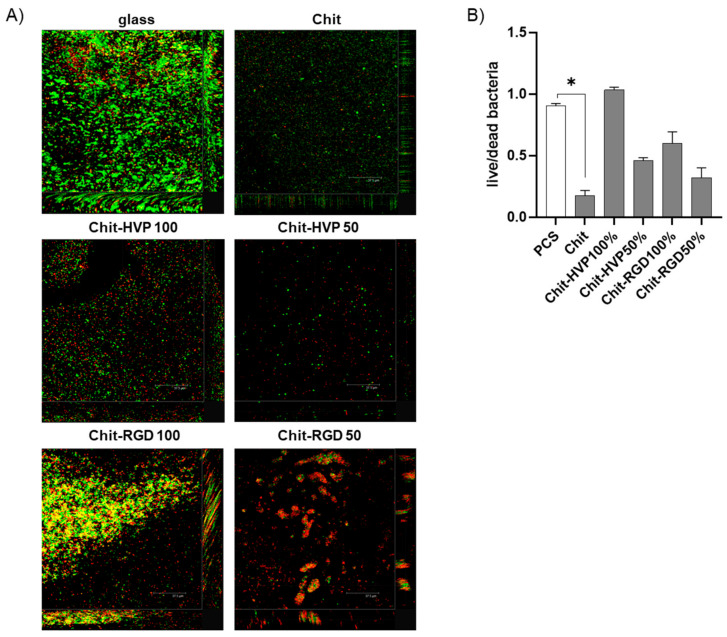
Cultures of *S. aureus* were grown on different functionalized chitosan matrices for 48 h, resulting in a mature biofilm in glass surfaces. Bacterial cultures were stained with a LIVE/DEAD BacLight Bacterial Viability Kit and (**A**) images were acquired using a Leica TCSNT/SP2 confocal microscope. Scale bar: 37.5 μm. The images are representative of three independent experiments, each performed in duplicate. (**B**) Data were analyzed using ImageJ and the ratio of live (bacteria with SYTO9 green fluorescence)/dead (bacteria with a high propidium iodide fluorescence and a low SYTO9 green fluorescence) bacteria was recorded. Data are reported as mean ± st err. * indicates *p* < 0.005 vs. glass surfaces. Chit: chitosan. % indicates the concentration of HVP or RGD.

**Table 1 ijms-22-05916-t001:** Oligonucleotide sequences.

Gene (Accession #)	Sequence	
*Gapdh* (NM_001289726)	Fw: 5′-agtgccagcctcgtcccgta-3′	Rv: 5′-caggcgcccaatacggccaa-3′
*Spp1* (NM_000582)	Fw: 5′-aagtttcgcagacctgacatc-3′	Rv: 5′-ggctgtcccaatcagaagg-3′
*Runx2* (NM_001024630)	Fw: 5′-cagtgacaccatgtcagcaa-3′	Rv: 5′-gctcacgtcgctcattttg-3′

## Data Availability

Not applicable.

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
