# Peer review of "Bio-Functionalized Chitosan for Bone Tissue Engineering"

_ijms, 2021, doi:10.3390/ijms22115916_

Round 1

Reviewer 1 Report

This experimental study is under the scope of this journal; the topic is relevant for readers, and this research deals with potentially significant knowledge to the field. 

  • However, there are some concerns about the present manuscript: 

Abstract

  • Need to be reformulated,
  • Descrived Aim, Metodology and conclusions.
  • In the results, is important to show more information, add some of the p-values.

Introduction  

  • Line 39-43 “Several papers reported on chitosan and its potential application in 

biomedical fields” this sentence need to be supported with more references.  In the literature the Chitosan have different application – on of these applications was Lyophilized chitosan scaffold, it used in the dentistry application. Please, read this article, Palma et al. (DOI: 10.1016/j.joen.2017.03.005) investigated in an animal study the usage of chitosan scaffolds for dental pulp regeneration (Regenerative Dentistry). This was added inside the root canal dentine walls to see recovery dental tissues, but it has seen an increase the bone regeneration. Please, read also this article https://doi.org/10.3390/polym12020436. 

Discussion 

  • Please, clarified what was the limitations of this study? —

-And also, clarified the future perspectives also add in the discussion. 

Conclusion

  • The conclusion section should more thoroughly summarize the results, this section is too long and too complex.

References

The titles of references have a different format, the title of the article is written in capital letters at the beginning of words, others only in lower case. Also, the standardized format of presentation in the journal's name. Because names have written in a different format, one is not abbreviated, others are not.

Author Response

We would like to thank Reviewer #1 for the revision of our Manuscript. We would like to respond in the following way to the points raised by the Reviewer:

-Abstract: We reformulated the abstract. All changes are underlined in the revised version of the Manuscript. 

- Introduction, Line 39-43: We thank the Reviewer for the suggestions. New reference were introduced in the INtroduction.

- Discussion: limitations and future prospectives of the study were added in the revised version of the Manuscript.  

-Conclusion: as suggested by the Reviewer, this section was shortened.

 -Reference: the reference style was homogenized. 

Reviewer 2 Report

I have read the manuscript titled "Biofunctionalized Chitosan for Bone Tissue Engineering" submitted to International Journal of Molecular Sciences" by Brun et al. Overall, its a nice work with sound experimental designs and conclusions supported by the results.

Minor comments -

  1. Normally, cell viability is assessed after 24 h of culture. Would authors explain why cell viability was assessed only 2 h of cell culture using MTT assay?
  2. For increased readability, authors should consider moving Figure 1-6 to supplementary information.
  3. Would authors expect to see similar antibacterial properties of Chitosan-HVP if they had used a Gram negative bacterium? If not, would reducing the % of HVP and increasing the % of RGD improve the properties. Please add this to the discussion.

Author Response

We thank Reviewer #2 for the revision of our Manuscript. We would like to respond to the points raised by the Reviewer in the following way:

- "Normally cell viability is assessed after 24 h...":  We agree with the Reviewer, usually cell viability is assessed after 24 hours of incubation with a drug to investigate toxicity. However, the aim of our experiment was to assess the biocompatibility of different functionalized chitosan and in particular, its ability in tethering osteoblast cells during the initial stages of cell-surface interaction providing us evidence of the immediate ability of functionalization in cell binding. We, therefore, set up the experiment to estimate the viable cells still attached to the tested material after 2 hours in culture. 

- "For increased readability...": We thank the Reviewer for his/her suggestion. However, we think that this study is interdisciplinary research where the chemical and analytical analysis supports the biological observations and vice versa. We therefore believe that both parts are equally important and we decided to keep all the figures in the main text. We hope the Reviewer will agree with us. 

- "Would authors expect to see...": We thank the Reviewer for this interesting suggestion. More considerations were added in the Discussion. All changes are underlined in the revised version of the Manuscript.    

Round 2

Reviewer 1 Report

This research is under the scope of this journal; the topic is interesting for readers and this research deals with potentially significant knowledge to the field and an open new way for future studies.

The authors improved the quality of the manuscript after the reviewer's indications.